# Parkin, as a Regulator, Participates in Arsenic Trioxide-Triggered Mitophagy in HeLa Cells

**Zhewen Zhang †, Juan Yi †, Bei Xie, Jing Chen, Xueyan Zhang, Li Wang, Jingyu Wang, Jinxia Hou and Hulai Wei \***

School of Basic Medical Sciences, Lanzhou University, Lanzhou 730000, China; zhangzhw@lzu.edu.cn (Z.Z.); yij@lzu.edu.cn (J.Y.); xieb@lzu.edu.cn (B.X.); chenjing@lzu.edu.cn (J.C.); zhangxy@lzu.edu.cn (X.Z.); wangli15@lzu.edu.cn (L.W.); wangjingyu@lzu.edu.cn (J.W.); houjx17@lzu.edu.cn (J.H.)
\* Correspondence: weihulai@lzu.edu.cn; Tel.: +86-13893158796
† These authors contribute equally to this work.

**Abstract:** Parkin is a well-established synergistic mediator of mitophagy in dysfunctional mitochondria. Mitochondria are the main target of arsenic trioxide (ATO) cytotoxicity, and the effect of mitophagy on ATO action remains unclear. In this study, we used stable Parkin-expressing (YFP-Parkin) and Parkin loss-of-function mutant (Parkin C431S) HeLa cell models to ascertain whether Parkin-mediated mitophagy participates in ATO-induced apoptosis/cell death. Our data showed that the overexpression of Parkin significantly sensitized HeLa cells to ATO-initiated proliferation inhibition and apoptosis; however, the mutation of Parkin C431S significantly weakened this Parkin-mediated responsiveness. Our further investigation found that ATO significantly downregulated two fusion proteins (Mfn1/2) and upregulated fission-related protein (Drp1). Autophagy was also activated as evidenced by the formation of autophagic vacuoles and mitophagosomes, increased expression of PINK1, and recruitment of Parkin to impaired mitochondria followed by their degradation, accompanied by the increased transformation of LC3-I to LC3-II, increased expression of Beclin1 and decreased expression of P62 in YFP-Parkin HeLa cells. Enhanced mitochondrial fragmentation and autophagy indicated that mitophagy was activated. Furthermore, during the process of mitophagy, the overproduction of ROS implied that ROS might represent a key factor that initiates mitophagy following Parkin recruitment to mitochondria. In conclusion, our findings indicate that Parkin is critically involved in ATO-triggered mitophagy and functions as a potential antiproliferative target in cancer cells.

**Keywords:** arsenic trioxide; Parkin; HeLa; apoptosis; autophagy



## 1. Introduction

Parkin is an E3 ubiquitin ligase involved in the elimination of damaged mitochondria [1–3]. Parkin is normally localized to the cytosol and translocated to depolarized mitochondria in a PTEN-induced kinase (PINK1)-dependent manner [4–6]. Damaged mitochondria accumulate PINK1 on their outer membrane, and after phosphorylation, recruit and activate Parkin to induce mitophagy [7,8]. Mitophagy is a specific autophagic process for the removal of damaged or depolarized mitochondria through the selective targeting of such mitochondria to the autophagic pathway [9]. It remains controversial whether mitophagy is pro-survival or pro-death during cancer therapies. Carroll et al. showed that wild-type Parkin was greatly sensitized toward apoptosis induced by mitochondrial depolarization but not by proapoptotic stimuli that failed to activate Parkin [10,11]. Johnson et al. provided evidence of a specific ubiquitin E3 ligase that might inactive Bax to promote cell survival [12–14]. Furthermore, the mechanism by which Parkin influences cell death remains to be elucidated.

Arsenic trioxide (ATO) is an effective agent for treating acute promyelocytic leukemia. Its anticancer activity has been supported by numerous studies [15–17]. Our previous

studies demonstrated that ATO leads to cancer cell death through different molecular mechanisms, including apoptosis, cell cycle arrest, autophagy, and excessive reactive oxygen species (ROS) [18–20]. Moreover, other studies found sufficient evidence for ATO-induced glutathione (GSH) level changes, DNA methyltransferase (DNMT) inhibition, nuclear factor kappa B (NF-κB) signaling pathway alterations, and so on [21]. Mechanistic studies of ATO in cancer cells are challenging, probably because of the complexity of the processes mediating its anticancer effects.

Mitochondria are one of the primary targets of ATO cytotoxicity [16,17]. Niu et al. indicated that ATO inhibited the proliferation of HepG2 cells by initiating mitophagy [22]. Watanabe et al. demonstrated that ATO induces the mitochondrial translocation of Parkin and ubiquitination of the voltage-dependent anion channel 1 (VDAC1) in HL-1 cardiomyocytes [23]. It has been shown that both mitophagy and the essential role of Parkin in mitochondrial quality control are critically involved in ATO-induced toxicity. Based on these reports, the present study was conducted to examine the role of Parkin in ATO-induced HeLa cell injury by observing its effects on mitophagy, mitochondrial function and apoptosis.

## 2. Materials and Methods

### 2.1. Cell Culture

Hela cells were obtained from the Experimental Center of School of Basic Medical, Lanzhou University. YFP-Parkin HeLa cells were retrieved from Hanming Shen's laboratory, and were originally provided as a gift from Dr. Richard Youle. Cells were cultured in Dulbecco's Modified Eagle Medium (DMEM) with 10% fetal calf serum and maintained in culture flasks at 37 °C in a humidified chamber containing 5% $CO_2$.

### 2.2. Plasmid Constructs

Parkin mutant (C431S) complementary DNAs (cDNAs) were generated by PCR using pEGFP-Parkin C431S (Addgene, 45877) as templates and inserted into pLVX-AcGFP1-N1 Vector (TaKaRa, 632154) with a C-terminal AcGFP1 protein (AcGFP1-Parkin C431S).

### 2.3. Establishment of the AcGFP1-Parkin Mutant (C431S) HeLa Stable Cell Lines

The three plasmids, AcGFP1-Parkin C431S, pMD2.G (Addgene, 12259) and psPAX2 (Addgene, 12260), were co-transfected into the virus packaging cell line 293T using Lipo2000. After 48 h and 72 h, the supernatant was harvested and concentrated. Then, HeLa cells were infected with the lentiviral supernatant expressing AcGFP1-Parkin C431S. After 48 h of infection, selection reagent puromycin (4 μg/mL) was added to the cells in fresh medium. After 3 days, the cells were reseeded at a low density with selection reagent (puromycin, 1 μg/mL) and incubated for 10 days. Lastly, single clones were expanded in a new dish with fresh medium.

### 2.4. Reagents and Antibodies

ATO (GB673-77) was purchased from Beijing Chemical (Beijing, China). DMEM and fetal bovine serum (FBS) were purchased from Hyclone (South Logan, UT, USA). The 3-(4,5-dimethylthiazol-2-yl)-2,5-diphenyl-2,5-tetrazolium bromide (MTT, M8180), dimethyl sulfoxide (DMSO, D8731), carbonyl cyanide m-chlorophenylhydrazone (CCCP, C6700) and Propidium Iodide (PI, C0080) were purchased from Solabio. Acetylcysteine (NAC, S1623) was purchased from Selleck. Annexin V-PE/7-AAD Apoptosis Detection Kit (40310ES20), MitoTracher®Red CMXRos (40741ES50) and 4′,6-diamidino-2-phenylindole (DAPI, 40728ES03) were from Yeasen. Fluorometric intracellular ROS Kit (MAK145) was from Sigma. Thr primary antibodies we used were Bcl-2 (Cell Signaling Technology,15071, 1:500 dilution), Bax (Cell Signaling Technology, 2772, 1:1000 dilution), GAPDH (Immunoway, YM3215, 1:5000 dilution), COX4 (GeneTex, GTX114330, 1:1000 dilution), Caspase-3 (Cell Signaling Technology, 9661, 1:500 dilution), Caspase-9 (Cell Signaling Technology, 9505, 1:500 dilution), Beclin 1 (Cell Signaling Technology, 3738, 1:1000 dilution),

P62 (GeneTex, GTX00955, 1:1000 dilution), LC3 (GeneTex, GTX100240, 1:1000 dilution), Parkin (Sigma, P5748, 1:1000 dilution), PINK1 (GeneTex, GTX107851, 1:1000 dilution), Drp1 (CST, CST5391, 1:1000 dilution), Mfn1 (CST, CST147395, 1:1000 dilution) and Mfn2 (CST, CST11925, 1:1000 dilution). Secondary antibodies were all from Immunoway and included anti-mouse-HRP (RS0001, 1:5000 dilution) and anti-rabbit-HRP (RS0002, 1:5000 dilution).

### 2.5. Cytotoxicity Assay

The MTT assay was carried out to evaluate the cytotoxicity of ATO. Briefly, cells ($5 \times 10^3$) were seeded in a 96-well plate and allowed to adhere overnight. Then, the cells were treated with different concentrations of ATO for 24 h and 48 h. A total of 10 μL of the MTT reagent (5 mg/mL) was added to each well. After an additional incubation for 4 h, 100 μL 10% SDS was added to dissolve formazan crystals. Absorbance values of 570 nm were recorded with a microplate reader (Bio-Tek, Winooski, VT, USA). All assays were repeated three times and data were presented as mean $\pm$ SD.

### 2.6. Colony Formation Assays

Cells were plated in 12-well plates ($1 \times 10^3$ cells per well) and treated with the indicated concentrations of ATO for 14 days. Then, cells were washed in PBS, fixed with 10% formaldehyde for 15 min and stained withed 1% crystal for 5 min before counting the number of colonies.

### 2.7. Analysis of Cell Cycle and Apoptosis

The distribution of the cell cycle was explored by staining DNA with Propidium Iodide (PI). After treatment, $1 \times 10^6$ cells were collected and fixed with 70% ethanol overnight. Cells were again washed with PBS and incubated at 37 °C for 30 min with RNase A (0.1 mg/mL) and PI (5 mg/mL). Then, cells' suspension was analyzed using flow cytometry (Beckman Coulter, Miami, FL, USA). The Annexin V-PE/7-AAD Apoptosis Detection Kit was used to determine whether ATO induced apoptosis in Hela cells. Similarly, after treatment, $1 \times 10^6$ cells were collected and suspended in binding buffer and stained with Annexin V/PE (5 μL) and 7-AAD (10 μL) at room temperature in the dark for 15 min. Cells undergoing apoptosis were analysed by flow cytometry. The apoptotic rate was determined for each condition as follows: Apoptotic rate = (early apoptotic rate +late apoptotic rate) $\times$ 100%.

### 2.8. Transmission Electron Microscopy

As described previously [24], cells (treated with ATO at 2.5 μmol/L or 5 μmol/L for 48 h) were fixed with 2.5% glutaraldehyde and 2% paraformaldehyde in 0.1 mol/L phosphate buffer, followed by 1% osmium tetroxide. After dehydration, thin sections were stained with uranyl acetate and lead citrate for observation. The ultrastructure of the cells was examined with a JEMI1230 transmission electron microscope (JEOL, Tokyo, Japan).

### 2.9. Mitochondrial Staining

Cells were grown on coverslips in six-well plates and cultured overnight. After treatment, the cells were washed with PBS thrice and added to 3.7% paraformaldehyde to stand for 10 min. Cells were then washed with PBS and stained with MitoTracher®Red CMXRos (100 nM) for 30 min or DAPI (5 μg/mL) for 15 min in the dark, at room temperature. Co-stained cells were washed and immediately observed using a fluorescence microscope (Olympus, Japan) equipped with a 20-objective lens. The area of red and green fluorescence was measured using software Image-Pro Plus 6.0.

### 2.10. Reactive Oxygen Species Detection

After treatment, the cells were rinsed with PBS and the reactive oxygen species (ROS) level was measured with the Fluorometric intracellular ROS Kit. The fluorescence mi-

croscope was used to observe the changes in ROS. Fluorescence intensity was analyzed quantitatively using Image-Pro Plus 6.0.

### 2.11. Western Blot Analysis

After treatment, cells were collected in a 70 μL RIPA buffer containing 1 mM PMSF (Beyotime, Haimen, China). The protein concentration was determined using the BCA kit (Solarbio, Bejing, China). Protein extracts (30 μg) were separated by 8–12% SDS-PAGE and electrophoretically transferred onto polyacrylamide difluoride membranes. After blocking with 5% non-fat milk in PBST for 1 h, the membranes were incubated with primary antibodies and secondary antibodies and visualized using ECL with film or CLINX Hemiscope (QinXiang, China). The relative signal intensity of bands was determined with Image J software. Protein expression levels were standardized by normalizing to GAPDH.

### 2.12. Statistical Analysis

Statistical analyses were performed using the SPSS16.0 software. All of the experiments were repeated three times in duplicate. The results were expressed as the mean $\pm$ SD. One-way ANOVA followed by Tukey's least significant difference post hoc test was used to analyze statistical differences between groups under different conditions. Statistical significance was set at $p < 0.05$.

## 3. Result

### 3.1. Parkin Aggravates the Proliferation Inhibition of HeLa Cells by ATO Treatment

To examine the function of Parkin in modulating cell proliferation, we used a HeLa cell line stably expressing YFP-Parkin to investigate the effects of Parkin on the viability of HeLa cells treated with ATO. The viability of the cells following treatment with various concentrations of ATO (0, 1, 2, 4, 6, and 8 μmol/L) for 24 h and 48 h was examined with MTT assays. As shown in Figure 1A, cell viability was markedly decreased in a dose- and time-dependent manner in both ATO-treated HeLa cells and YFP-Parkin HeLa cells, and the $IC_{50}$ value of ATO in YFP-Parkin HeLa cells was much lower than that in HeLa cells (Figure 1B). The examination of the colony formation ability of the cells revealed, in the same way, that the overexpression of YFP-Parkin significantly decreased the colony number of YFP-Parkin HeLa cells compared to that of control HeLa cells after ATO administration (Figure 1C,D).

To further explore whether Parkin participates in the ATO-induced proliferation suppression of HeLa cells, we constructed a HeLa cell line with a loss-of-function muta-tion in Parkin (Parkin C431S). The mutation of Parkin C431S significantly weakened the proliferation-inhibitory effects of ATO on the YFP-Parkin HeLa cells, which were restored to close to the level of the control HeLa cells, as shown by MTT colorimetry and colony formation assays (Figure 1E–G). These findings indicate that the overexpression of Parkin could promote, in part, the sensitivity of HeLa cells to ATO.

### 3.2. Parkin Promotes ATO-Induced Apoptosis of HeLa Cells

To examine whether Parkin is a regulator of apoptosis in response to ATO treatment, the cells were double-stained with Annexin V-PE/7-AAD and PI. As shown in Figure 2A,B, ATO showed a much stronger apoptosis-inducing effect in YFP-Parkin HeLa cells than in HeLa cells, and the mutation of Parkin to Parkin C431S markedly impaired the apoptosis of ATO-treated YFP-Parkin HeLa cells. As an example, after treatment with 5 μmol/L ATO for 24 h, the apoptosis rates of HeLa cells, YFP-Parkin HeLa cells and Parkin C431S cells were $13.15 \pm 1.67\%$, $27.48 \pm 1.9\%$ and $21.01 \pm 3.1\%$, respectively. In addition, the analysis of the cell cycle distribution in ATO-treated HeLa cells showed that the overexpression of YFP-Parkin had no significant effects on the ATO-induced cell cycle changes that manifested as G2/M arrest (Figure 2C). As shown in Figure 2D, the expression of apoptosis-related proteins revealed that ATO produced a dose- and time-dependent increase in Cleaved-Caspase-9, Cleaved-Caspase-3 and Bax and a decrease in Bcl-2 proteins in both HeLa

and YFP-Parkin HeLa cells, and the latter was much more pronounced. All of the above evidence indicated that Parkin expression greatly improved the sensitivity of HeLa cells to ATO-stimulated apoptosis.

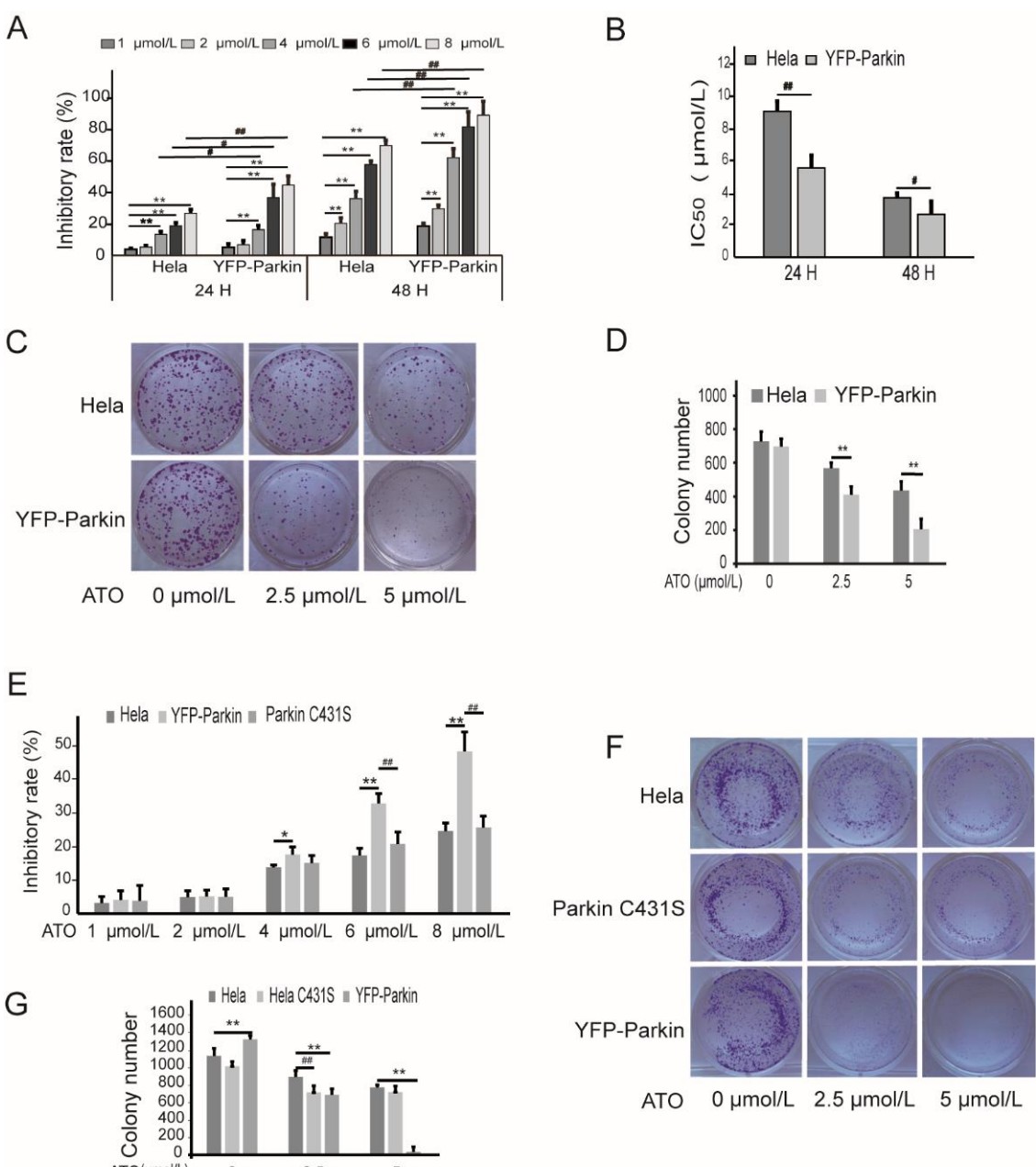

**Figure 1.** ATO modulates proliferation of HeLa cells. (**A**) MTT assays revealed that overexpression of YFP-Parkin significantly decreased the growth rate of the indicated cells. (**B**) The percentage of IC$_{50}$. (**C**) YFP-Parkin decreased the mean colony number in the colony formation assay. (**D**) Quantification of C. (**E**) MTT assays revealed that Parkin C431S rescued the growth rate after ATO treatment for 24 h. (**F**) Parkin C431S rescued the mean colony number in the colony formation assay. (**G**) Quantification of F. The histograms represent the mean ± SD of triplicate experiments. * $p < 0.05$ vs. 0 μmol/L group, ** $p < 0.01$ vs. 0 μmol/L group; # $p < 0.05$ vs. without Parkin group, ## $p < 0.01$ vs. without Parkin group.

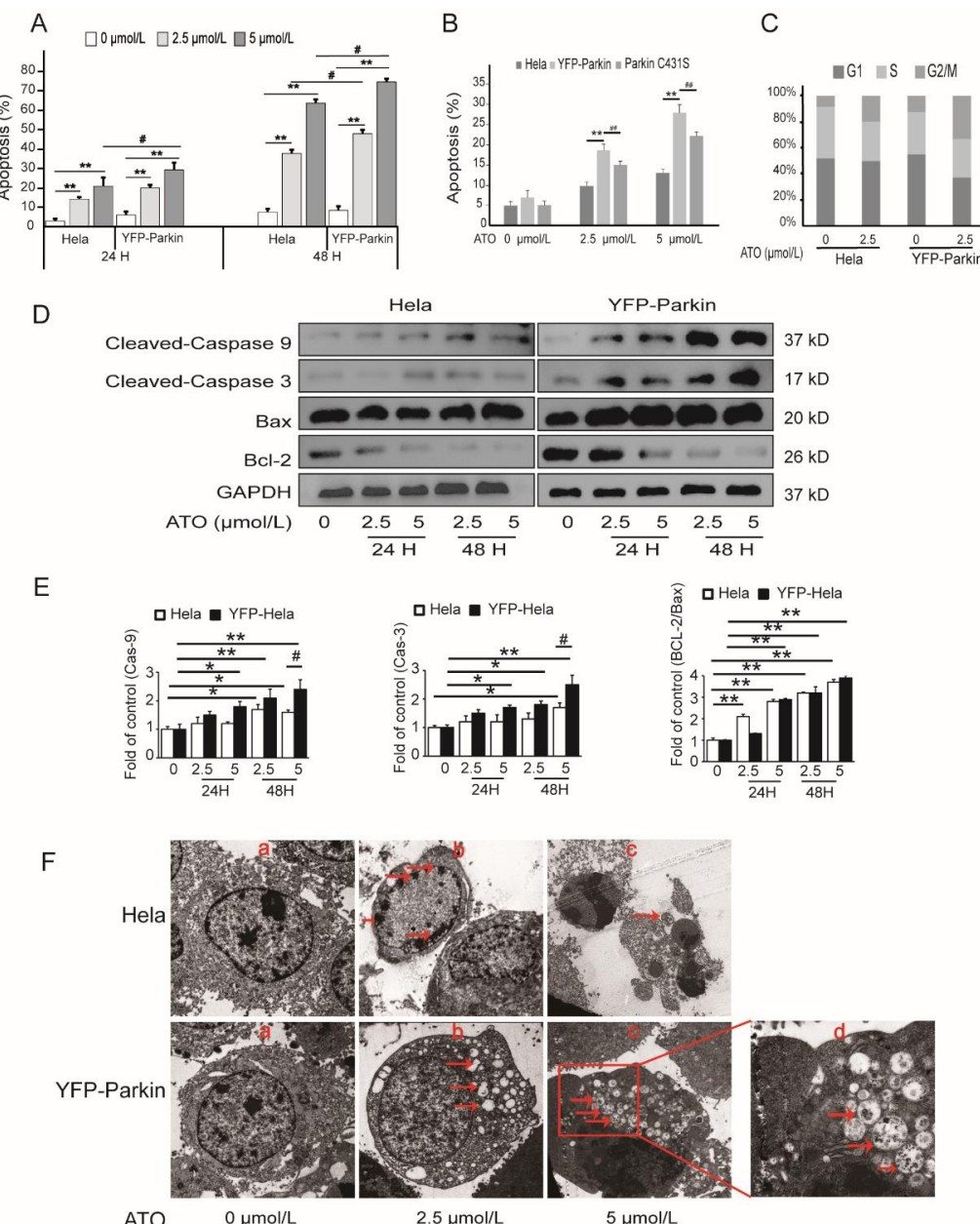

**Figure 2.** ATO induces the apoptosis of HeLa cells. (**A**) The apoptotic rate. (**B**) Parkin C431S rescued the apoptosis rate after ATO treatment for 24 h. (**C**) The percentage of cells in each phase of the cell cycle was analyzed after ATO treatment for 24 h by flow cytometry. (**D**) Western blot analysis of Cleaved-Caspase-3, Cleaved-Caspase-9, Bax, and Bcl-2 expression. (**E**) Quantification of D. (**F**) Morphology of apoptosis and autophagy was observed under a transmission electron microscope after ATO treatment for 24 h (a, b and c 5000×; d 15,000×). The arrowhead (↓) indicates the chromatin condensation, apoptotic bodies, autophagic vacuoles and mitophagosomes in apoptotic cells. The histograms represent the mean ± SD of triplicate experiments. * *p* < 0.05 vs. 0 µmol/L group, ** *p* < 0.01 vs. 0 µmol/L group, # *p* <0.05 vs. without Parkin group; ## *p* <0.01 vs. without Parkin group; GAPDH served as an internal control.

Transmission electron microscopy revealed that, as shown in Figure 2F, after exposure to ATO, both HeLa cells and YFP-Parkin HeLa cells exhibited morphological changes typical of apoptosis features, such as shrinkage and apoptotic bodies, which were more obvious in YFP-Parkin HeLa cells. At the same time, we also observed a large number of autophagic vacuoles and mitophagosomes in the cytoplasm of ATO-exposed YFP-Parkin

HeLa cells. These phenomena mean that Parkin strengthened ATO-triggered apoptosis in HeLa cells, which was likely achieved via the induction of autophagy or mitophagy.

### 3.3. PINK1/Parkin Pathway Was Involved in the ATO-Induced Mitochondrial Damage and Mitophagy

To ascertain whether the PINK1/Parkin pathway mediated mitophagy in the apoptosis of ATO-treated HeLa cells, we investigated the effects of Parkin on ATO-induced mitochondrial dysfunction in YFP-Parkin HeLa cells. To assess whether ATO exposure disturbed mitochondrial fission and fusion, we detected the expressions of fission and fusion-related proteins in both HeLa and YFP-Parkin HeLa cells. Our results showed that the expression of Drp1 increased, and the expression of Mfn1/2 decreased in the ATO-treated cells compared to the control cells (Figure 3A–D). We also observed the decreased expression of COX IV, a common mitochondrial marker (Figure 3E,F), and reduced Mito-Tracker fluorescence of mitochondria (Figure 4A–D) in ATO-treated HeLa and YFP-Parkin HeLa cells, which suggested that ATO might cause mitochondrial damage.

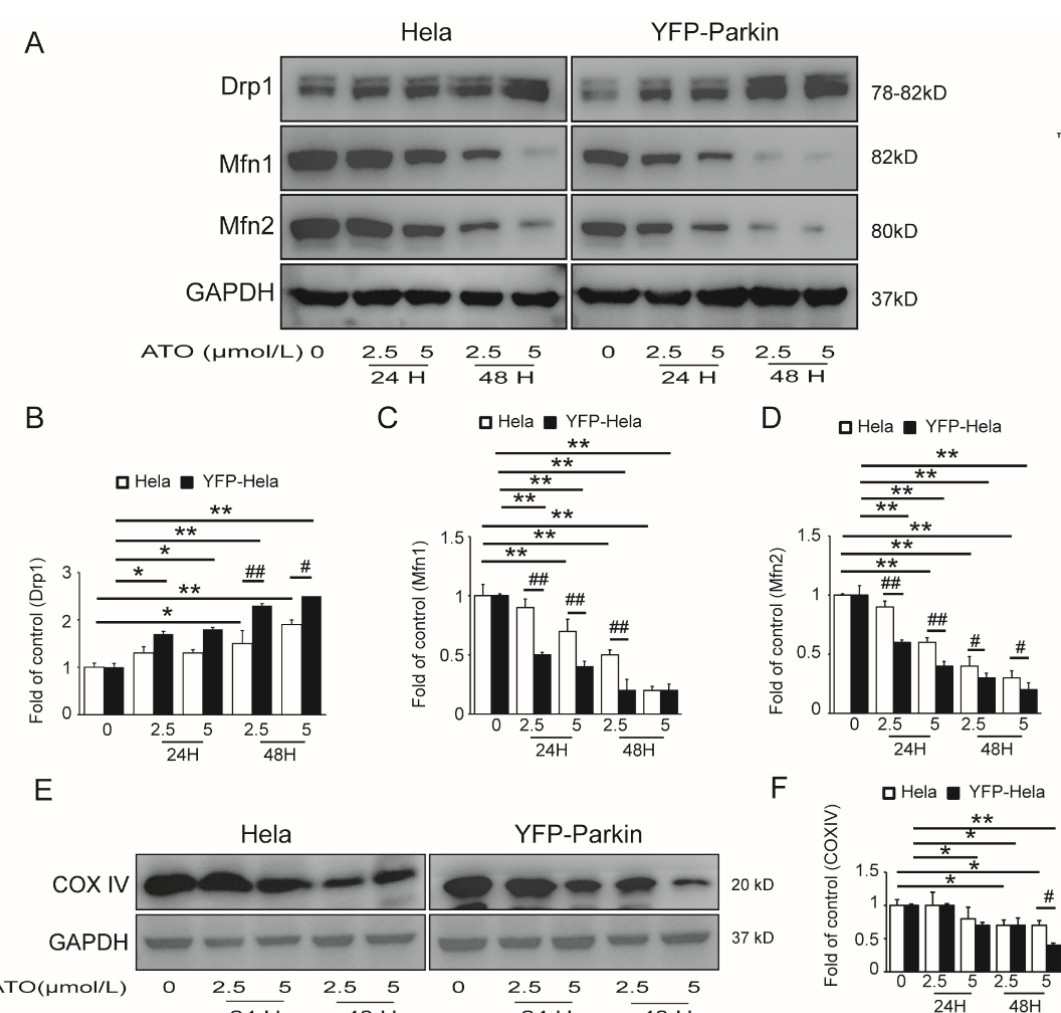

**Figure 3.** Effects of ATO exposure on mitochondrial dynamics in HeLa cells. (**A**) Western blot analysis of Drp1, Mfn1 and Mfn2 expression. (**B–D**) Quantification of A. (**E**) Western blot analysis of COX IV expression in the indicated cells. (**F**) Quantification of E. The histograms represent the mean $\pm$ SD of triplicate experiments. * $p < 0.05$ vs. 0 $\mu$mol/L group, ** $p < 0.01$ vs. 0 $\mu$mol/L group, # $p < 0.05$ vs. without Parkin group, ## $p < 0.01$ vs. without Parkin group; GAPDH served as an internal control.

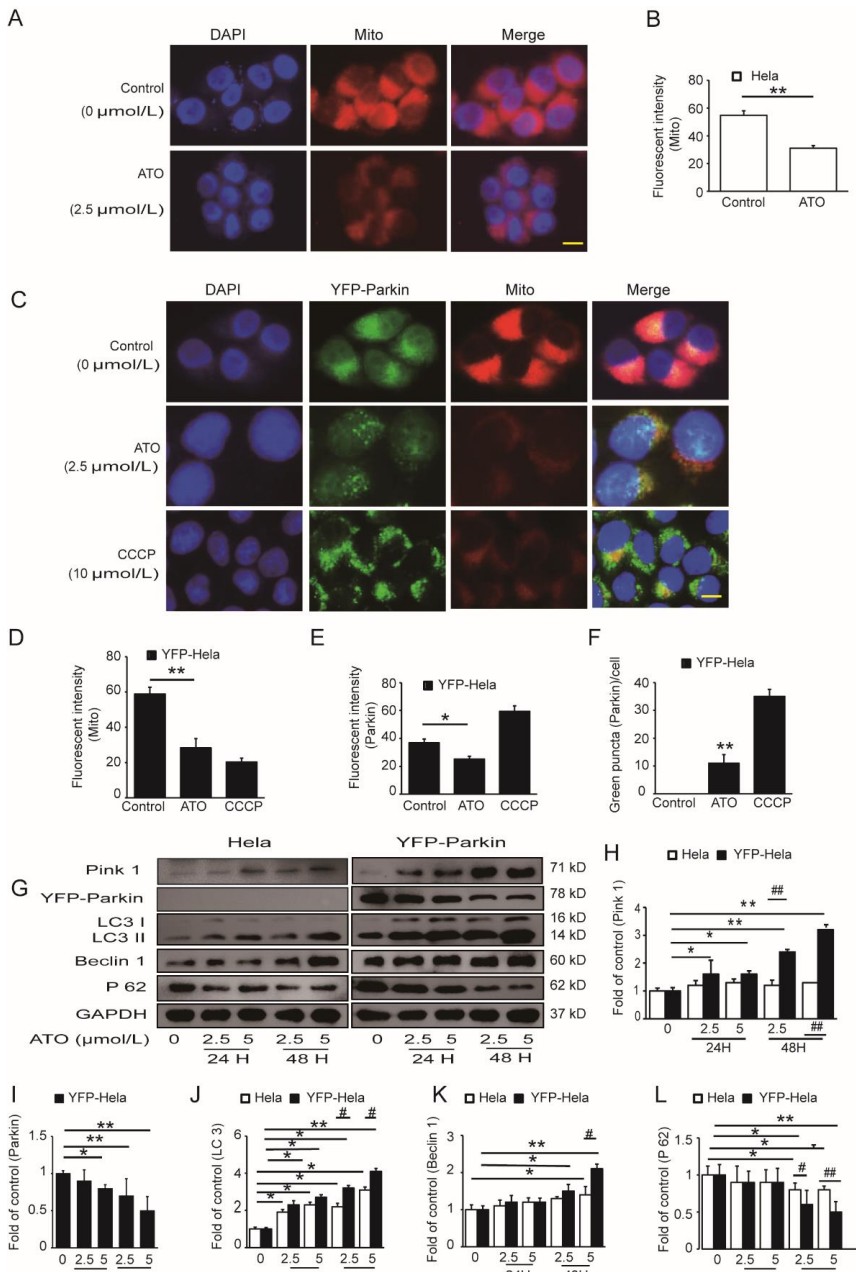

**Figure 4.** Regulation of PINK1/Parkin-mediated mitophagy by ATO. (**A**) HeLa cells were fixed and immunostained for nuclei (blue) and mitochondria (red) after ATO treatment for 24 h. The red fluorescence indicated the normal mitochondrial membrane potential. The samples were analyzed using a fluorescence microscopy. Scale bar, 10 μm. (**B**) shows the quantification of the red fluorescence, which was performed using Image J software. (**C**) YFP-Parkin HeLa cells were fixed and immunostained for nuclei (blue) and mitochondria (red) after ATO treatment for 24 h. The red fluorescence indicated the normal mitochondrial membrane potential. The green fluorescence was the marker for YFP-Parkin proteins. The level of mitophagy was evaluated by the number of green puncta. The samples were analyzed using a fluorescence microscopy. Scale bar, 10 μm. (**D**) Quantification of the red fluorescence, which was performed using Image J software. (**E**) Quantification of the green fluorescence, which was performed using Image J software. (**F**) Quantification of green puncta, which was performed using Image J software. (**G**) Western blot analysis of Parkin, PINK1, LC3, Beclin 1 and P62 expression in the indicated cells. (**H**–**L**) Quantification of G. The histograms represent the mean ± SD of triplicate experiments. * $p < 0.05$ vs. 0 μmol/L group, ** $p < 0.01$ vs. 0 μmol/L group, # $p < 0.05$ vs. without Parkin group; ## $p < 0.01$ vs. without Parkin group; GAPDH served as an internal control.

Usually, Parkin is recruited by PINK 1 to damaged mitochondria and then they are degraded via the activation of mitophagy. To test whether mitochondria labeled by Parkin display decreased membrane potential, we pulsed the cells with MitoTracker red, a potentiometric mitochondrial dye, before fixation. YFP-Parkin selectively accumulated on those mitochondria with MitoTracker staining (Figure 4C–F). These results also indicated that ATO resulted in mitochondrial damage and that Parkin might be recruited to impaired mitochondria to trigger their degradation by PINK1/Parkin-mediated mitophagy. To confirm this possibility, we assessed the expression of PINK1 and Parkin, as well as the autophagy-related proteins LC3, Beclin1 and P62, in ATO-exposed YFP-Parkin HeLa cells. The results of the Western blotting examination showed that after treatment with 2.5 and 5 μmol/L ATO for 24 h and 48 h, the increased expression of PINK1 and decreased expression of Parkin were accompanied by the increased transformation of LC3-I to LC3-II, increased expression of Beclin1 and decreased expression of P62 in YFP-Parkin HeLa cells compared to control HeLa cells (Figure 4G–L). These findings provide extremely strong evidence that Parkin is critically involved in the mitophagy of HeLa cells in response to ATO.

### 3.4. ROS Activate the PINK1/Parkin Pathway to Mediate Mitophagy and Mitophagic Death in ATO-Exposed HeLa Cells

Intracellular ROS production is one of the antitumoral mechanisms of ATO. To investigate the effect of the overexpression of Parkin on ATO-simulated ROS generation and whether ROS initiate the Parkin-mediated mitophagic death of HeLa cells by ATO, we incubated YFP-Parkin HeLa cells with 2.5 μmol/L ATO for 6 h. As shown in Figure 5, intracellular ROS overproduction (Figure 5A,B) occurred, accompanied by mitochondrial damage (Figure 5D–G), Parkin degradation and enhanced apoptosis (Figure 5C), and all of these effects could be reversed by the ROS scavenger acetylcysteine (NAC) (Figure 5D–I). These data suggested that the generation of ROS gave rise to mitochondrial dysfunction and activated the PINK1/Parkin pathway to trigger mitophagy and mediate mitophagic apoptosis in ATO-treated HeLa cells.

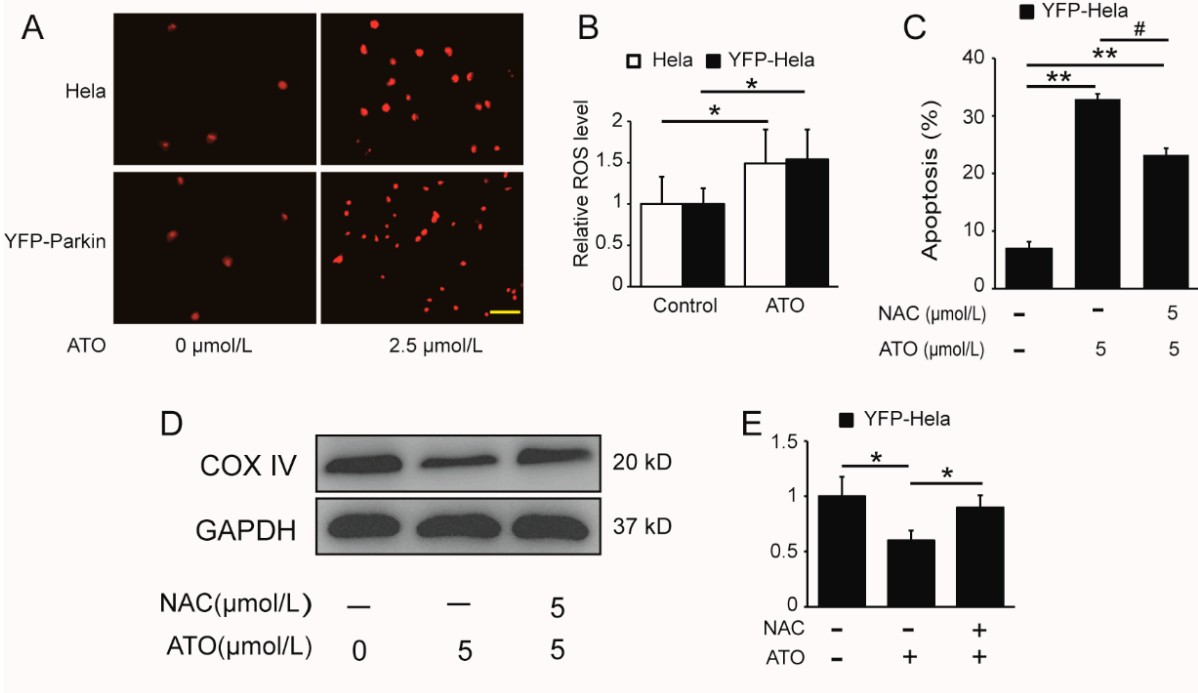

**Figure 5.** *Cont.*

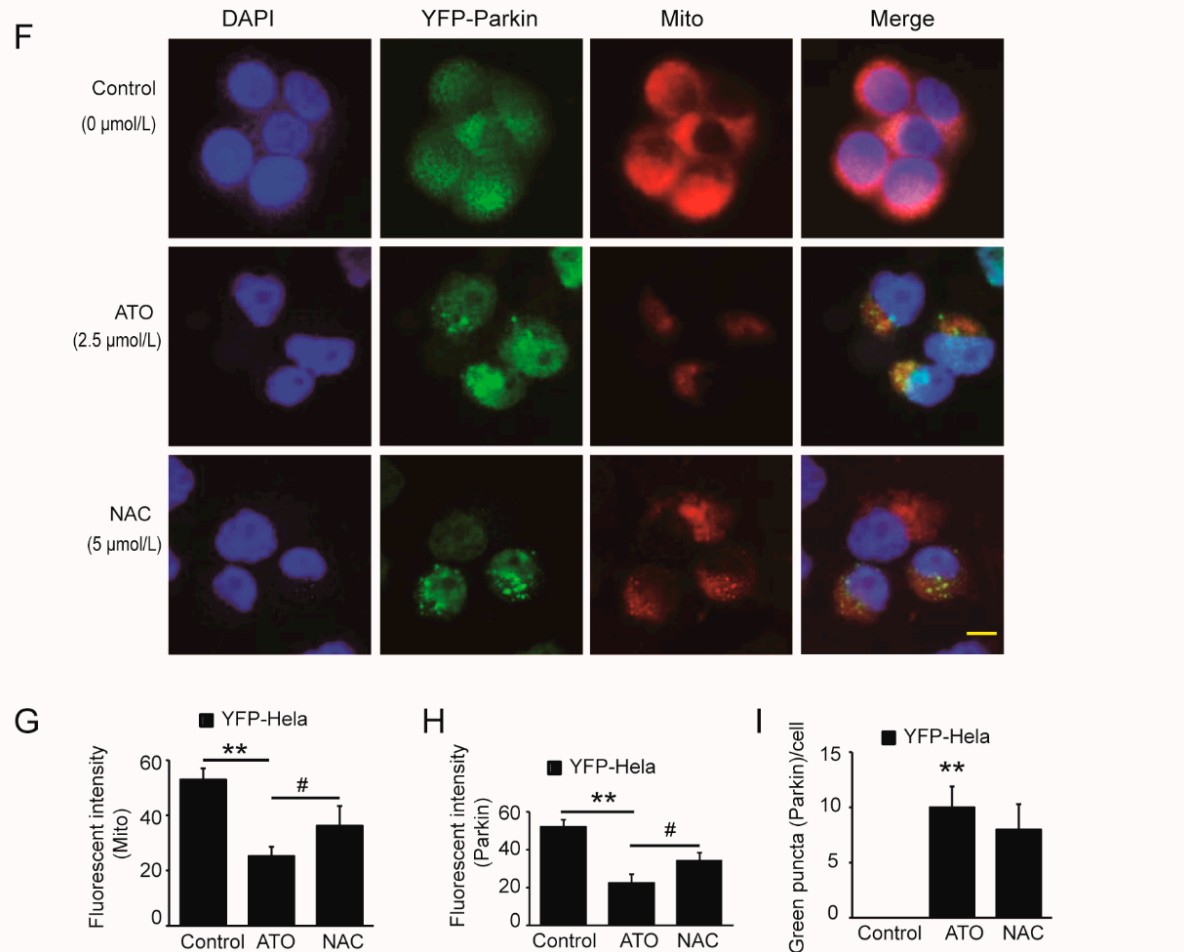

**Figure 5.** Potentiating effect of ATO on ROS production. (**A**) Red fluorescence indicates the production of ROS after cell treatment for 6 h, and the images were analyzed using a fluorescence microscopy. Scale bar, 10 μm. (**B**) shows the quantification of the red fluorescence, which was performed using Image J software. (**C**) YFP-Parkin HeLa cells were treated with the indicated concentrations of ATO for 24 h before detection of cell apoptosis by flow cytometry. (**D**) Western blot analysis of COX IV expression rescued after ATO treatment for 24 h in YFP-Parkin HeLa cells. (**E**) Quantification of D. (**F**) YFP-Parkin HeLa cells were fixed and immunostained for nuclei (blue) and mitochondria (red) after ATO treatment for 24 h. The red fluorescence indicated the normal mitochondrial membrane potential. The green fluorescence was the marker of YFP-Parkin proteins. The level of mitophagy was evaluated by the number of green puncta. The samples were analyzed using a fluorescence microscopy. Scale bar, 10 μm. (**G**) Quantification of the red fluorescence, which was performed using Image J software. (**H**) Quantification of the green fluorescence, which was performed using Image J software. (**I**) Quantification of green puncta, which was performed using Image J software. The histograms represent the mean ± S.D. of triplicate experiments. * $p < 0.05$ vs. 0 μmol/L group, ** $p < 0.01$ vs. 0 μmol/L group, # $p < 0.05$ vs. without Parkin group. Without NAC group; GAPDH served as an internal control.

## 4. Discussion

Parkin plays an essential role in mitochondrial quality control and mitochondria are one of the primary targets of ATO cytotoxicity [17,25,26]. To date, few studies have directly investigated the changes in Parkin in response to ATO cytotoxicity. In this study, we investigated the possible role of Parkin in promoting mitophagy in YFP-Parkin HeLa cells in response to ATO. Our data showed that the overexpression of Parkin significantly inhibited cell viability and induced mitochondrial damage as well as apoptosis in HeLa cells. During this process, ATO induced PINK1/Parkin-mediated mitophagy. Furthermore,

we demonstrated that ROS may represent a key factor that promotes mitophagy following Parkin recruitment to mitochondria.

The molecular mechanism of ATO-induced apoptosis in tumor cells has been widely reported [16,17,21]. ATO activates different signaling pathways in different tumor cells. In this study, ATO caused mitochondrial impairment, including cytochrome c release and subsequent caspase-3 and -9 activation. ATO induced apoptosis via a mitochondria-dependent pathway in HeLa cells. The typical morphological characteristics of apoptosis were observed by electron microscopy after treatment with ATO. We generated HeLa cells stably expressing YFP-Parkin and found that Parkin overexpression can significantly increase the apoptosis of HeLa cells and inhibit cell proliferation. A large number of autophagic vacuoles and mitophagosomes appeared in the cytoplasm in YFP-Parkin HeLa cells exposed to ATO. These results suggested that ATO-induced mitochondrial damage activated PINK1/Parkin-mediated mitophagy. Furthermore, the mutation of the Parkin C431S alleviated ATO-induced cell apoptosis and proliferation. These findings showed that Parkin plays an important role in the activation of mitophagy under ATO treatment.

Previous studies indicated that the PINK1/Parkin pathway was widely accepted as the chief mechanism of mitophagy, closely related to the regulation of mitochondrial membrane potential, mitochondrial fusion/fission and mitochondrial function [27]. Narendra et al. demonstrated that Parkin is selectively recruited to dysfunctional mitochondria in mammalian cells and that after recruitment, Parkin mediates the engulfment of mitochondria by autophagosomes and their subsequent degradation [7]. Okatsu et al. showed that the mitophagy mechanism is controlled by PINK1/Parkin [28]. We hypothesized that ATO increased the clearance of impaired mitochondria by activating Parkin-mediated mitophagy. The Western blot results showed a significant reduction in Mfn1/2 in cells. The immunofluorescence assays showed that the intensity of the red fluorescence clearly decreased by the ATO treatment of the two cell lines. In YFP-Parkin HeLa cells, the intensity of the red fluorescence was much lower than that in HeLa cells. We next determined whether Parkin was recruited to mitochondria. As shown in Figure 4C, we found that YFP-Parkin colocalized with mitochondria with lower MitoTracker staining. These data indicate that the completion of mitophagy requires the recruitment of Parkin to damaged mitochondria post-ATO treatment, which is consistent with a previous study. PINK1 is upstream of Parkin and functions to recruit Parkin to mitochondria to trigger the process of mitophagy. The Western blot analysis also showed that the level of PINK1 was markedly increased in YFP-Parkin HeLa cells after treatment with ATO compared with HeLa control cells. These results suggested that the PINK1/Parkin pathway was activated in YFP-Parkin HeLa cells by ATO.

Subsequent experiments revealed that the ROS content was increased in ATO-induced cells. Xu et al. demonstrated that ROS functions as an upstream signal in the PINK1/Parkin pathway to mediate mitophagy progression [29]. Xiao et al. confirmed that ROS promotes mitophagy following Parkin translocation to mitochondria [30]. Our results are similar to those conclusions. ROS was clearly observed as early as 6 h post-ATO treatment. A marked overlay of Parkin and mitochondria could be observed 24 h after adding ATO in our study. Considering that ROS is upstream activators of Parkin, we determined whether ROS is involved in Parkin regulation under ATO. As shown in Figure 4D, after pretreatment with NAC for 1 h, exposure to ATO not only enhanced the green (Parkin) and red (mitochondria) fluorescence but also increased the visible colocalization of Parkin and mitochondria. It is speculated that the execution of mitophagy may be mainly propelled by ROS. Consistent with this, the protein expression of COX IV and the apoptosis ratio were markedly rescued by NAC compared with cells treated with ATO alone. Our data indicate that ROS acts downstream of Parkin recruitment to impaired mitochondria to drive mitophagy forward, while the activation of the PINK1/Parkin signaling pathway may contribute to the completion of mitophagy.

## 5. Conclusions

In conclusion, we found that cell viability decreased, the apoptotic rate increased, and PINK1/Parkin-mediated mitophagy was activated in YFP-Parkin-overexpressing cells treated with ATO. Further investigation indicated that ROS might represent a key factor that promotes mitophagy following Parkin recruitment to mitochondria. Our findings indicate that Parkin is critically involved in ATO-triggered mitophagy and it functions as a potential antiproliferative target in cancer cells.

**Author Contributions:** Conceptualization, H.W.and Z.Z.; methodology, J.Y., B.X. and Z.Z.; software, L.W.; J.C., J.W. and J.H. performed the research; Z.Z., X.Z. and J.Y. analyzed the data; H.W. and Z.Z. wrote the paper. All authors have read and agreed to the published version of the manuscript.

**Funding:** This work was supported by the Fundamental Research Funds for the Central Universities (No. lzujbky-2017-138); the Natural Science Foundation of Gansu Province (No. 18JR3RA291; No. 20JR5RA290; No. 17JR5A214; No. 21JR7RA448).

**Data Availability Statement:** Not applicable.

**Conflicts of Interest:** The authors declare no conflict of interest.

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
