# Peer review of "Parkin, as a Regulator, Participates in Arsenic Trioxide-Triggered Mitophagy in HeLa Cells"

_cimb, doi:10.3390/cimb44060189_

Round 1
Reviewer 1 Report
Comments and Suggestions for Authors
Zhewen et al in their manuscript ‘Parkin, as a regulator, participates in arsenic trioxide-triggered mitophagy in HeLa cells’ wants to demonstrate the role of Parkin in the mitophagy activated by ATO treatment in HeLa cells.
The topic may be interesting but authors have to do other experiments to make strengthen their conclusion.
Major point:
- It would be interesting seeing all the experiments (in particular immunofluorescence staining and ROS) after the transfection of mutated Parkin (C431S). This point would strengthen the conclusions.
- It’s not clear for me why the authors decided to do all the immunofluorescence staining (that are fundamental for their conclusion) at 2,5 umol/L ATO concentration after 24 hrs of treatment when themselves demonstrate that this combination is not very effective on mitophagy ( COX IV fig 3A). I would like to see the results at different time and different doses of ATO (Figure 3 and 4).
- Figure 2: I would like to know if authors studied cell cycle distribution also at different doses of ATO, in a time dependent manner and after transfection of mutated Parkin. Can be interesting studied the cell cycle distribution which can change whit the different states of apoptosis.
- It would be better if the authors studied ROS whit FACS analysis also at different time and different doses and whit mutated Parkin because FACS analysis gives quantitative results.
Minor points:
- Authors don’t explain what is CCCP and why they used it.
- In all the immunofluorescence figures quantization and bars are missing
- Abbreviations are missing (for example line 34, 49, 55,…). Authors have to add all the abbreviations.
- Superscripts are missing in all materials and methods (line 65, 96, 104, 110, 111, 114).
Author Response
Dear Editors and Reviewers:
Thank you for your letter and for the reviewers’ comments concerning our manuscript entitled “Parkin, as a regulator, participates in arsenic trioxide-triggered mitophagy in HeLa cells” (cimb-1635152). Those comments are all valuable and very helpful for revising and improving our paper, as well as the important guiding significance to our researches. We have studied comments carefully and have made correction which we hope meet with approval. Revised portion are marked in red in the paper. The main corrections in the paper and the responds to the reviewer’s comments are as flowing:
Major point:
- It would be interesting seeing all the experiments (in particular immunofluorescence staining and ROS) after the transfection of mutated Parkin (C431S). This point would strengthen the conclusions.
Thank you very much for pointing out the defects of our manuscript! Indeed, in our experiment the mutated Parkin (C431S) was used as a control to confirm Parkin was involved in ATO effects, and in the subsequent experiment we mainly focused on the reaction of Parkin-Hela, some observational index, such as ROS and Parkin, was not observed in Parkin mutant (C431S) cells. This is a defect in our study. In the ongoing studies we will overcome the deficiency according to Reviewer’s direction.
- It’s not clear for me why the authors decided to do all the immunofluorescence staining (that are fundamental for their conclusion) at 2,5 umol/L ATO concentration after 24 hrs of treatment when themselves demonstrate that this combination is not very effective on mitophagy ( COX IV fig 3A). I would like to see the results at different time and different doses of ATO (Figure 3 and 4).
Thank the reviewer for the valuable guidance. In our study we treated firstly the cells with different concentration of ATO at different time, and then the appropriate concentration of ATO, i.e., 2.5 and 5umol/L, and suitable time were chosed for the subsequent experiment. According to the reviewer’s comment, in this short time of revising, for us it is almost impossible to re-do additional experiments of different time and different doses of ATO in Fig 3 and Fig 4. And in fact, this is not necessarily indispensable for the conclusion of Fig 3 and Fig 4 in our manuscript.
- Figure 2: I would like to know if authors studied cell cycle distribution also at different doses of ATO, in a time dependent manner and after transfection of mutated Parkin. Can be interesting studied the cell cycle distribution which can change whit the different states of apoptosis.
Thank you for your suggestion. Indeed, in this manuscript we didn’t observe the ATO-induced cell cycle distribution in Parkin-mutated HeLa cells because of its only slight changes in Parkin-HeLa cells. Lots of studies have confirmed the relationship of cell cycle distribution and apoptosis, and it is an indisputable fact that the cell cycle distribution may change with the different states of apoptosis. Following the reviewer's advices we may focus on this interesting study in our future item.
- It would be better if the authors studied ROS whit FACS analysis also at different time and different doses and whit mutated Parkin because FACS analysis gives quantitative results.
Thanks very much for your valuable suggestion. For detection of ROS the FACS analysis indeed is a better assay because of its both qualitative and quantitative property. But in our study, in order to observe the intracellular location and distribution of ROS, the changes of ROS were examined with Fluorometric intracellular ROS Kit by fluorescence microscope, which, in fact, might be similarly quantified using Image-Pro Plus 6.0 and we have done so.
Minor points:
- Authors don’t explain what is CCCP and why they used it.
Thanks for your direction and sorry for negligence. CCCP is the abbreviation of Carbonyl cyanide m-chlorophenydrazone, which is frequently used as a prototypical mitophagy inducer or mitochondrial uncoupler. In our study CCCP was used to test whether mitochondrial depolarization causes Parkin accumulation on mitochondria. In the revised manuscript, we have replenished the information about CCCP in Line 225, which was also marked with Reference 7.
- In all the immunofluorescence figures quantization and bars are missing
Thanks for your reminding. In the revised manuscript, in all the fluorescence figures the fluorescence intensity was quantified and the data were rearranged and analyzed statistically in added new Figures.
- Abbreviations are missing (for example line 34, 49, 55,…). Authors have to add all the abbreviations.
Thank you very much for your kindness. In the revised manuscript, we have carefully checked and added the abbreviations in the full text.
- Superscripts are missing in all materials and methods (line 65, 96, 104, 110, 111, 114).
Sorry for our carelessness. In the revised manuscript, we have carefully checked the full text, and added all the missed superscripts, especially the Lines 65, 96, 104, 110, 111 and 114, in materials and methods section.
We tried our best to improve the manuscript and made some changes in the manuscript. We appreciate for Editors/Reviewers’ warm work earnestly, and hope that the correction will meet with approval.
Once again, thank you very much for your comments and suggestions.
Thank you and best regards.
Yours sincerely,
Corresponding author: Wei Hulai
E-mail: weihulai@lzu.edu.cn

Reviewer 2 Report
Comments and Suggestions for Authors
This manuscript describes in vitro data about how parkin overexpression regulates mitophagy and subsequent cell death in arsenic trioxide-treated HeLa cells. The Authors also provide mechanistic data to explain the described observations. The data presented here is clinically relevant; however, the description of experiments and visualization of the data requires amendments.
Specific suggestions:
- Is mitophagic cell death (lanes 23 and 249) an accepted term? If not, please, revise.
- In lane 120: “As described previously”, a reference is missing.
- Parkin immunostaining is not described in 2.9. MitoTracker staining is not an immunostaining, please, revise.
- In 2.11., working dilutions of antibodies should be provided.
- With respect to Figures 1C-G and 2C-D, the length of the treatment(s) should be described in the corresponding legends.
- In Figures 2D, 3A, 3D, and 4D, size markers should be provided. Moreover, how the densitometry results, displayed below the blot images, have been obtained is not described. Has the densitometry data been analyzed statistically?
- The arrowhead(s) in Figure 2E are not properly visible.
- There is no direct evidence for the statement in the title of 3.3.
- In Figures 3B-C, 4A, and 4C, size markers should be provided. The images should be quantified to draw conclusions (e.g. MitoTracker intensity, co-localization of MitoTracker and Parkin signals).
- All abbreviations should be given when first mentioned in the text.
- There are minor grammatical or spelling errors in lanes 14, 62, 96, 104, 110, 114, and 128.
Author Response
Dear Editors and Reviewers:
Thank you for your letter and for the reviewers’ comments concerning our manuscript entitled “Parkin, as a regulator, participates in arsenic trioxide-triggered mitophagy in HeLa cells” (cimb-1635152). Those comments are all valuable and very helpful for revising and improving our paper, as well as the important guiding significance to our researches. We have studied comments carefully and have made correction which we hope meet with approval. Revised portion are marked in red in the paper. The main corrections in the paper and the responds to the reviewer’s comments are as flowing:
Responds to the reviewer’s comments:
Reviewer #2:
- Is mitophagic cell death (lanes 23 and 249) an accepted term? If not, please, revise.
Thank you very much for your positive comment for our manuscript, and sorry for the emerging term we used, mitophagic cell death, confusing you. In fact, the term mitophagic cell death is a universal accepted paradigm of cell demise, which has been regularly used in many papers by other authors (Park H, et al. Parkin Promotes Mitophagic Cell Death in Adult Hippocampal Neural Stem Cells Following Insulin Withdrawal. Front Mol Neurosci. 2019;12:46; Meyer N, et al. AT 101 induces early mitochondrial dysfunction and HMOX1 (heme oxygenase 1) to trigger mitophagic cell death in glioma cells. Autophagy. 2018;14(10):1693-1709; Kim EH, Choi KS. A critical role of superoxide anion in selenite-induced mitophagic cell death. Autophagy. 2008; 4(1):76-8., etc)
- In lane 120: “As described previously”, a reference is missing.
Sorry for our inattention. We have revised this error, and replenished a reference (Line 126, Reference 24).
- Parkin immunostaining is not described in 2.9. MitoTracker staining is not an immunostaining, please, revise.
Thanks for your suggestion. Indeed, in this manuscript, Parkin plasmid with YFP gene was transfected into HaLa cells, which produce Parkin protein labelled with YPF and displayed fluorescence under fluorescent microscope so the fluorescence of Parkin in the cells is not produced by immunofluorescent staining. Also, MitoTracker staining is not an immunostaining, MitoTracker is a potentiometric mitochondrial dye, by which mitochondria can labelled. So, the title of “2.9 Immunostaining assay” is not appropriate, and according to your comment, we have revised the 2.9 as “Mitochondrial staining”.
- In 2.11., working dilutions of antibodies should be provided.
Sorry for our carelessness. We have provided the working dilutions of each antibodies (Line 92-100).
- With respect to Figures 1C-G and 2C-D, the length of the treatment(s) should be described in the corresponding legends.
Thanks very much for your direction. In the legends of Figures 1C-G and Figures 2C-D, we have replenished the description of the length of the ATO treatment (Lines 111, 183 and 214)
- In Figures 2D, 3A, 3D, and 4D, size markers should be provided. Moreover, how the densitometry results, displayed below the blot images, have been obtained is not described. Has the densitometry data been analyzed statistically?
Thanks very much and sorry for our poor description. According to reviewer’s suggestion, we have added the size markers in the corresponding position in Figures 2D, 3A, 3D, and 4D. In Lines 151-152, in 2.11 section, we provided a supplemental description how the densitometry results were obtained, and all the data were rearranged and analyzed statistically as in Figs 2E, 3A, 3I-N, 4E.
- The arrowhead(s) in Figure 2E are not properly visible.
Thank your very much. According to your comment and in order to improve the visible of the arrowheads in Figure 2E, we have changed the small arrowheads as big red arrowheads in Fig 2.
- There is no direct evidence for the statement in the title of 3.3.
- Thanks for the valuable guidance and the valuable suggestion. Indeed, based on our actual data there were no enough direct evidence to support the expression of “PINK1/Parkin are recruited to ATO-induced damaged mitochondria to trigger mitophagy”, but the present data is enough to support the statement that PINK1/Parkin pathway was involved in the ATO-induced mitochondrial damage and mitophagy. To accurately state the implication of Result 3.3, in the revised manuscript, we changed the title of 3.3 as “PINK1/Parkin pathway was involved in the ATO-induced mitochondrial damage and mitophagy”.
- In Figures 3B-C, 4A, and 4C, size markers should be provided. The images should be quantified to draw conclusions (e.g. MitoTracker intensity, co-localization of MitoTracker and Parkin signals).
Thank you for your direction. According to reviewer’s suggestion, we have added the size markers in the corresponding position in Figures 3B-C, 4A, and 4C. And we have also quantified the MitoTracker intensity, co-localization of MitoTracker and Parkin, etc., which was displayed in Figs 3D, 3F-H and 4G-H.
- All abbreviations should be given when first mentioned in the text.
Thank you very much. In the full text, we have carefully corrected all the abbreviations and each abbreviation was given the full title when first mentioned.
- There are minor grammatical or spelling errors in lanes 14, 62, 96, 104, 110, 114, and 128
Thanks very much for your critical comments. We have carefully checked and corrected the all the grammatical and spelling errors in the text, especially the Lines 14, 62, 96, 104, 110, 114 and 128.
We tried our best to improve the manuscript and made some changes in the manuscript. We appreciate for Editors/Reviewers’ warm work earnestly, and hope that the correction will meet with approval.
Once again, thank you very much for your comments and suggestions.
Thank you and best regards.
Yours sincerely,
Corresponding author: Wei Hulai
E-mail: weihulai@lzu.edu.cn

Reviewer 3 Report
Comments and Suggestions for Authors
The manuscript, "Parkin, as a regulator, participates in arsenic trioxide-triggered 2 mitophagy in HeLa cells" is a well written manuscript. experimental description and result are well described. Authors has checked ATO induced mitophagy through parkin and studied all related parameters including LC3, cleaved caspase, Bcl2-Bax. The missing part here is fission fusion of mitochondria. They need to show more mitochondrial structural/functional parameters as is already well established that Parkin recruitment to mitochondria is important for induction of mitophagy, thus here they are only establishing that ATO can induced mitophagy. It would be better they can confirm more mitochondria related changes (fission-fusion etc).
Author Response
Thanks very much for your direction. According to reviewer’s suggestion, we have conducted the supplementary experiments. Mitochondrial fission and fusion are governed by several key proteins that maintain mitochondrial quality control. To assess whether ATO exposure disturbed mitochondrial fission and fusion, we detected the expression of protein levels of Drp1, Mfn1 and Mfn2. The changes in these proteins were showed in Fig 3.

Round 2
Reviewer 1 Report
Comments and Suggestions for Authors
The manuscript appears to be improved compared to the previous time, especially in fluorescence images that now are significative.
But I disagree with authors when they say ' this is not necessarily indispensable for the conclusion of Fig 3 and Fig 4 in our manuscript.' All the conclusion of the paper are done on incorrect dose of ATO because, as authors shown in figures 2D, 3A, 3I and also 4D, the low dose of ATO is not very effective on apoptosis/mitophagy processes. The results given by western blot and by IFA do not correlate.
The authors have to try to produce some immunofluorescence results with higher dose of ATO to be sure about the results.
Author Response
Dear Editors and Reviewers:
Thank you for your letter and for the reviewers’ comments concerning our manuscript entitled “Parkin, as a regulator, participates in arsenic trioxide-triggered mitophagy in HeLa cells” (cimb-1635152). Those comments are all valuable and very helpful for revising and improving our paper, as well as the important guiding significance to our researches. We have studied comments carefully and have made correction which we hope meet with approval. The main responds to the reviewer’s comments are as flowing:
Responds to the reviewer’s comments:
Reviewer #1:
The manuscript appears to be improved compared to the previous time, especially in fluorescence images that now are significative.
But I disagree with authors when they say ' this is not necessarily indispensable for the conclusion of Fig 3 and Fig 4 in our manuscript.' All the conclusion of the paper are done on incorrect dose of ATO because, as authors shown in figures 2D, 3A, 3I and also 4D, the low dose of ATO is not very effective on apoptosis/mitophagy processes. The results given by western blot and by IFA do not correlate.
The authors have to try to produce some immunofluorescence results with higher dose of ATO to be sure about the results.
Thank you very much for your patience and kindness to re-review our revised manuscript, and thanks again for your valuable guidance and suggestion. In our study, to choose the appropriate concentration and treating time of ATO, we treated HeLa cells with different concentration of ATO at different time, because HeLa cells were relatively sensitive to ATO, when treatment with higher concentration of ATO ( >5 µmol/L), lots of the cells were injured and fall off the flask or the coverslips, and the dying cells floated in a culture media, which might interfere with the results of fluorescent staining. So we had to select a lower and relatively effective concentration of ATO to do fluorescent staining. In fact, the results of WB were from all the cultured cells (adherent cells + floating cells, i.e., live cells + dying cells), and the results of fluorescent staining only from the adherent cells on the coverslips (i.e., live cells), which was why there were some difference in results between WB and IFA. According to your question and suggestion, we re-did some assays, with higher concentration of ATO, in Figs 3 and 4, and unfortunately, as shown in the attached figures, most of the cells were dying and floated in the media after treated with higher concentration of ATO, which were unsuitable and difficult to do fluorescent staining. For the above reasons, choice of lower concentration of ATO, in the fluorescent staining assays in Figs 3 & 4, was the last resort, and in fact, the results of the fluorescent staining assays, in Figs 3 & 4, were still somewhat meaningful although the lower concentration of ATO is not very effective.
Thank you again for your laborious reviewing work and constructive suggestion.
Attached figures
We tried our best to improve the manuscript and made some changes in the manuscript. We appreciate for Editors/Reviewers’ warm work earnestly, and hope that the correction will meet with approval.
Once again, thank you very much for your comments and suggestions.
Thank you and best regards.
Yours sincerely,
Corresponding author: Wei Hulai
E-mail: weihulai@lzu.edu.cn

Reviewer 2 Report
Comments and Suggestions for Authors
The manuscript was substantially improved. Therefore, I suggest the current version of this manuscript for publication.
Author Response
Dear Editors and Reviewers:
Thank you for your letter and for the reviewers’ comments concerning our manuscript entitled “Parkin, as a regulator, participates in arsenic trioxide-triggered mitophagy in HeLa cells” (cimb-1635152).
Thanks very much for your kind work and consideration on publication of our paper. On behalf of my co-authors, we would like to express our great appreciation to reviewer.
Once again, thank you very much for your comments and suggestions.
Thank you and best regards.
Thank you and best regards.
Yours sincerely,
Corresponding author: Wei Hulai
E-mail: weihulai@lzu.edu.cn
Round 3
Reviewer 1 Report
Comments and Suggestions for Authors
I am sorry for the technical problems encountered by the authors but the problems can be solved. For example the authors can be evaluate WB analysis at low doses of ATO dividing death/live cells.
I'm still thinking that the experiments requested above are critical for the strength of the work.
Author Response
Thanks very much for your sympathy and kindness. According to your suggestion, we have done our best to supplement some experimental data and improve our manuscript, and we hope genuinely to meet your comment and gain your understanding. Your suggestion is of great significance to our future researching works, and we will try our best to improve our experimental works, on the basis of your guidance, in the ongoing and future studies.
Thank you again for your kindness.